# Mesenchymal stem cell suppresses the efficacy of CAR-T toward killing lymphoma cells by modulating the microenvironment through stanniocalcin-1

**Rui Zhang[1†], Qingxi Liu[2,3†], Sa Zhou[4], Hongpeng He[4], Mingfeng Zhao[1]\*, Wenjian Ma[3,4]\***

[1]Department of Hematology, Tianjin First Central Hospital, School of Medicine, Nankai University, Tianjin, China; [2]State Key Laboratory of Medicinal Chemical Biology and College of Life Sciences, Nankai University, Tianjin, China; [3]Qilu Institute of Technology, Shandong, China; [4]College of Biotechnology, Tianjin University of Science and Technology, Tianjin, China

**\*For correspondence:**
mingfengzhao@sina.com (MZ);
ma_wj@tust.edu.cn (WM)

[†]These authors contributed equally to this work

**Competing interest:** The authors declare that no competing interests exist.

**Abstract** Stem cells play critical roles both in the development of cancer and therapy resistance. Although mesenchymal stem cells (MSCs) can actively migrate to tumor sites, their impact on chimeric antigen receptor modified T cell (CAR-T) immunotherapy has been little addressed. Using an in vitro cell co-culture model including lymphoma cells and macrophages, here we report that CAR-T cell-mediated cytotoxicity was significantly inhibited in the presence of MSCs. MSCs caused an increase of CD4$^+$ T cells and Treg cells but a decrease of CD8$^+$ T cells. In addition, MSCs stimulated the expression of indoleamine 2,3-dioxygenase and programmed cell death-ligand 1 which contributes to the immune-suppressive function of tumors. Moreover, MSCs suppressed key components of the NLRP3 inflammasome by modulating mitochondrial reactive oxygen species release. Interestingly, all these suppressive events hindering CAR-T efficacy could be abrogated if the stanniocalcin-1 (STC1) gene, which encodes the glycoprotein hormone STC-1, was knockdown in MSC. Using xenograft mice, we confirmed that CAR-T function could also be inhibited by MSC in vivo, and STC1 played a critical role. These data revealed a novel function of MSC and STC-1 in suppressing CAR-T efficacy, which should be considered in cancer therapy and may also have potential applications in controlling the toxicity arising from the excessive immune response.

## Editor's evaluation

This study uncovers the contributions of MSC on modulating CAR T-cell behaviour. Based on the importance in basic biology and its immediate impact on translational potential, all reviewers are satisfied on the advances in this study.

## Introduction

Advances in chimeric antigen receptor modified T cell therapy (CAR-T) in recent years have shown enormous promise in cancer immunotherapy, which has produced unprecedented clinical outcomes, most notably for patients with hematologic malignancies (*Singh et al., 2016*; *Park et al., 2018*). Despite the striking achievements, CAR-T therapy is also facing many challenges such as the treatment-related severe toxicity and side effects, including cytokine release syndrome (CRS) and neurotoxicity (*Hong et al., 2020*; *Freyer and Porter, 2020*). CRS is the most common acute toxicity associated

**eLife digest** Immunotherapy is a type of cancer treatment that helps the immune system fight cancer. For example, chimeric antigen receptor T cell (CAR-T) therapy is used to target several types of blood cancer. It works by reprogramming patients' immune cells to target specific tumor cells. In blood cancers, CAR-T therapy works very well, but it can cause extreme responses from the patient's immune system, which can be life threatening. In solid tumors, CAR-T therapy is much less successful because the tumors secrete molecules into the space surrounding them, which weaken the immune processes that attack cancerous cells.

Stem cells are the master cells of the body. Originating in the bone marrow, they can repair and regenerate the body's cells. Cancer stem cells play a role in resistance to CAR-T therapy, due – in part – to their ability to renew themselves, but the role of another type of stem cell, called mesenchymal stem cells, was less clear. Mesenchymal stem cells develop into tissues that line organs and blood vessels. Although it is known that mesenchymal stem cells are present in most cancers and play a role in shaping and influencing the space around tumors, their impact on CAR-T therapy has not been studied in depth.

To find out more, Zhang et al. looked at the influence of a protein, called staniocalcin-1 (STC1), on CAR-T therapy, by studying cells grown in the laboratory and human tumor cells that had been implanted in mice.

Zhang et al. found that mesenchymal stem cells reduce the ability of CAR-T therapy to destroy cancer cells and that they needed STC1 to do this successfully. They also increased the expression of molecules that dampen the immune system, and suppressed molecules called inflammasomes, which are an important part of the way the immune system detects disease. Moreover, reducing the amount of STC1 that mesenchymal stem cells expressed restored the effectivity of CAR-T therapy.

This study increases our understanding of the way that mesenchymal stem cells affect CAR-T therapy. It has the potential to open up a new way of improving the efficiency of this treatment and of reducing the harmful side effects that it can cause.

---

with an excessive immune response that causes fever, hypotension, and respiratory insufficiency. The neurotoxicity induced by CAR-T therapy exhibits a diverse array of neurologic symptoms such as tremors, expressive aphasia, and impaired attention. The precise mechanism that causes these life-threatening side effects remains unclear (*Freyer and Porter, 2020*; *Jiang et al., 2019*). On the other hand, the success of CAR-T therapy in treating solid tumors is still very limited (*Martinez and Moon, 2019*). Identifying hurdles and potential mechanisms that impede the function of CAR-T cells is of vital importance to expanding its use. The immunosuppressive tumor microenvironment (TME) is one of the obstacles that diminishes the efficacy of CAR-T therapy, especially for solid tumors.

Among the many factors that can modulate TME and immune response, the impact of mesen-chymal stem cell (MSC) on CAR-T therapy has been little studied. MSC is a type of adult stem cell with high proliferative activity and multidirectional differentiation capacity. However, MSCs have additional paracrine effects that are believed to underlie their therapeutic functions (*Jiang and Xu, 2020*). By secreting a variety of cytokines into the tissue microenvironment, it has been known that MSCs can modulate extracellular matrix, promote angiogenesis, and suppress inflammation and apoptosis (*Keating, 2012*; *Wang et al., 2014*; *Regmi et al., 2019*). Some MSC-secreted cytokines, such as stromal cell-derived factor 1 and stem cell factor, play important roles in hematopoietic and immune regulation (*Kawaguchi et al., 2019*; *Markov et al., 2007*). In addition, studies suggest that MSCs can modulate the function of monocytic lineages cells, especially macrophages (*Németh et al., 2009*; *Ylöstalo et al., 2012*; *Choi et al., 2011*). Some reports also showed that MSCs could directly affect the functionality and cellular responses of T cells, Tregs, and memory T cells (*Cen et al., 2019*; *Tumangelova-Yuzeir et al., 2019*; *Luque-Campos et al., 2019*).

It was reported that human mesenchymal stem cells (hMSCs) could be activated by lipopolysac-charide (LPS)-stimulated macrophages to increase the expression and secretion of stanniocalcin-1 (STC1) (*Oh et al., 2014*). STC1 was a mitochondria-related glycoprotein originally identified as a calcium/phosphate regulating hormone in bony fishes, and later on, it was found to be a pleio-tropic factor involved in various degenerative diseases such as ocular and renal disease, as well as

idiopathic pulmonary fibrosis (*Yeung et al., 2012*; *Ohkouchi et al., 2015*). STC1 could improve the cell survival and regeneration of MSCs in a paracrine fashion (*Ono et al., 2015*). There was also evidence suggesting that STC1 played an oncogenic role in various types of tumors (*Du et al., 2011*; *Liu et al., 2010*). Based on a retrospective study of ~1500 clinical samples, it was concluded that high STC1 expression is associated with the poor clinical outcome of breast cancer (*Chang et al., 2015*). It was proved that STC1 is involved in several oxidative and cancer-related signaling pathways, such as NF-κB, extracellular-signal-regulated kinase (ERK), and c-Jun NH(2)-terminal kinase (JNK) pathways (*Nguyen et al., 2009*; *Chan et al., 2017*). The expression and secretion of STC1 in cancer tissue can be stimulated by external stimuli, including external cytokines and oxidative stress (*Nguyen et al., 2009*). Under hypoxia conditions, STC1 could be modulated by Hypoxia-inducible factor-1 (HIF-1) to facilitate the reprogramming of tumor metabolism from oxidative to glycolytic metabolism (*Yeung et al., 2005*). STC1 was also reported to participate in the process of epithelial-to-mesenchymal transition, which is associated with tumor invasion and the reshape of the tumor microenvironment, as well as increasing therapy resistance (*Pastushenko and Blanpain, 2019*).

Considering the pleiotropic role of STC1, especially its intercellular linkage between MSCs, cancer cells, and macrophage stimulation, it is interesting to know what role it plays in connection to the functions of MSC in TME. Therefore, we generated a stable STC1 knockdown MSC cell line. With a cell co-culture model containing CAR-T cells, hMSCs, macrophages, and Pfeiffer lymphoma cells to partially mimic the tumor microenvironment together with a xenograft mice model, here we studied the impacts of MSC on CAR-T efficacy and the potential immune response change in the presence and absence of STC1.

## Results

### Stable knockdown of STC1 in hMSC-inhibited cell migration, slightly suppressed cell proliferation, but no increase in apoptosis

To study the function of STC1, we first generated a stable knockdown cell line by lentivirus-based shRNA for the STC1 gene, and the expression of STC1 protein was evaluated by Western blot (*Figure 1A*). STC1 stable knockdown in hMSCs exhibited a minor effect in cell survival (*Figure 1B*) and slightly reduced proliferation rate based on the small increase in the proportion of cells in G0/G1 phases versus that in the S phase (*Figure 1C*) as determined by MTT (3-[4,5-dimethylthiazol-2 -yl]-2,5 diphenyl tetrazolium bromide) and Fluorescence-activated Cell Sorting (FACS) analysis. To investigate whether knockdown of STC1 affects cell migration, wound healing and transwell chamber assays were performed. After creating a 'scratch' in a monolayer of hMSCs, the closure of the gap was determined after 24 hr. As shown in *Figure 1D*, compared to control hMSCs, the gap was less filled in hMSC$^{shSTC1}$. The inhibitory effect on cell migration was further confirmed by a transwell assay. As shown in *Figure 1E*, there were significant migration and invasion observed in hMSCs$^{shCtrl}$, whereas there was a >30% reduction in migration across the transwell chamber membrane in hMSCs$^{shSTC1}$. To further determine whether knockdown of STC1 may have any lethal effect, apoptosis was determined by two different assays. To measure the early apoptosis, cells were stained with the Alexa Fluor 488 annexin V and the propidium iodide (PI) followed by flow cytometry to detect apoptosis-associated phosphatidylserine (PS) expression and membrane permeability (*Figure 1F*). Parallelly, no DNA fragmentation was detected as determined with the TUNEL assay (*Figure 1G*, the green dots were from the background due to overexposure). Both studies showed that knockdown of STC1 did not cause apoptosis of hMSCs.

### The presence of hMSCs inhibited CAR-T cell killing activity, but knockdown of STC1 completely abrogated this inhibition

To investigate the impact of hMSCs on CAR-T treatment, we used an in vitro cell co-culture model modified according to previous studies to mimic a simplified situation of tumor environment (*Singh et al., 2017*; *Liu et al., 2021*). The co-culture contained CD19 CAR-T cells, Pfeiffer cells that were from human diffuse large cell lymphoma, and M2 macrophages (derived from THP-1 cells by phorbol-12-myristate-13-acetate [PMA] polarization for 24 hr) at a cell number ratio of 1:3:1. The cell-killing activity of CAR-T cells toward Pfeiffer cells was determined by lactate dehydrogenase (LDH) cytotoxicity assay on total cell co-culture. As shown in *Figure 2A*, 67% of Pfeiffer cells were killed after being

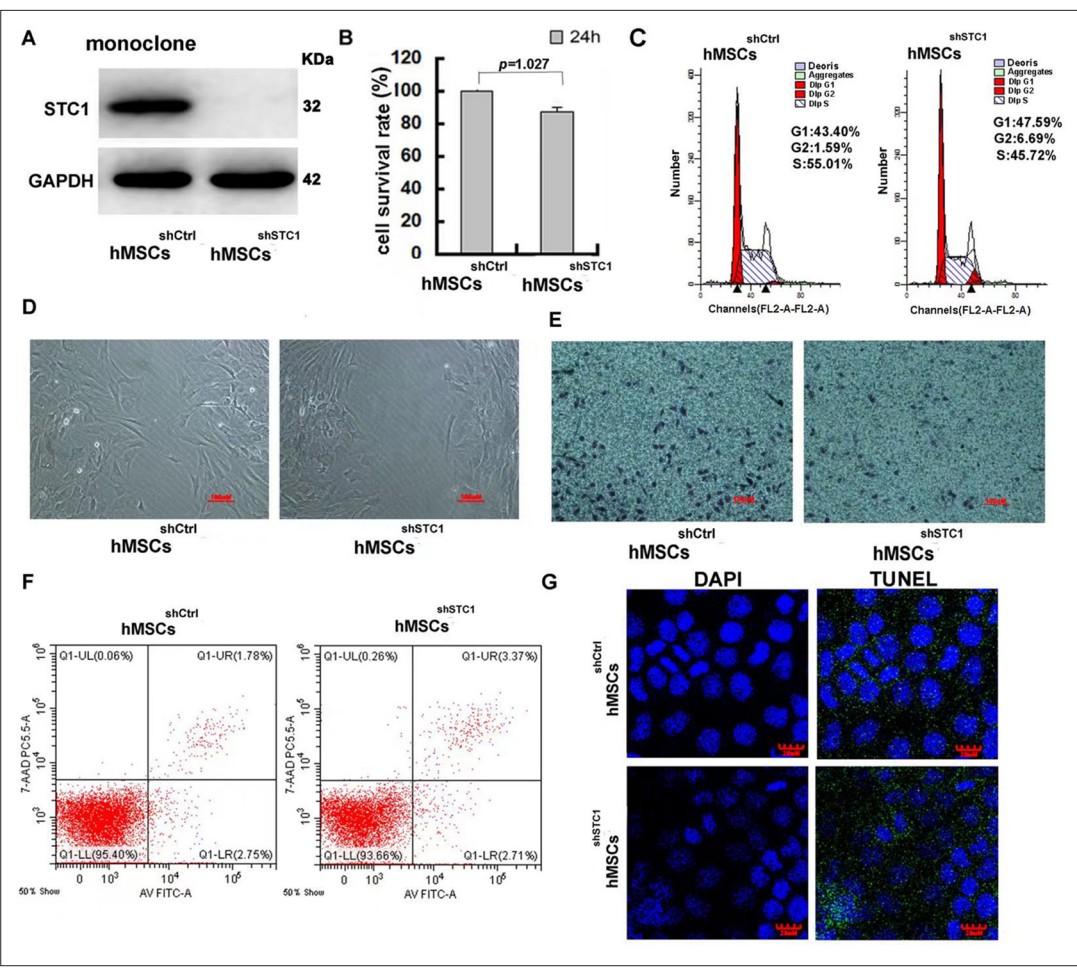

**Figure 1.** The impact of stanniocalcin-1 (STC1) knockdown on cell proliferation, migration, and apoptosis of hMSCs. (**A**) Western blot analysis of STC1 protein expression in hMSCs. (**B**) Cell viability determined by MTT, measurements are shown as the mean ± SD from three independent experiments. (**C**) FACS analysis of cell cycle progression on hMSCs w/o STC1 knockdown. (**D, E**) Knockdown of STC1 suppressed cell migration as determined by wound healing and transwell chamber assays. (**F**) Apoptosis determination by the Alexa Fluor 488 annexin V and PI detection. (**G**) DNA fragmentation determination by transferase-mediated dUTP nick-end labeling (TUNEL) assay.

The online version of this article includes the following source data for figure 1:

**Source data 1.** Labeled original blots of *Figure 1A*.

**Source data 2.** Unlabeled original blots of *Figure 1A*.

**Source data 3.** *Figure 1B* in Excel file.

exposed to CAR-T cells for 24 hr, and 93% were killed at 48 hr as compared to mock-treated control. After adding hMSCs into the co-culture, the cell-killing activity of CAR-T was significantly inhibited (*Figure 2A*). The number of hMSC added was the same as the CAR-T cell. Interestingly, the inhibitory effect of hMSCs on CAR-T cytotoxicity could be completely abrogated if knockdown STC1 gene in hMSCs. These results for the first time revealed that CAR-T efficacy could be affected by the presence of MSCs, and the gene STC1 played a critical role.

## Co-culturing with hMSCs caused an increase of CD4⁺ T cells and Treg cells but a decrease of CD8⁺ T cells

Previous studies have demonstrated that the composition of CD4⁺ and CD8⁺ T cell subsets was crucial for CAR-T cell efficacy (*Sommermeyer et al., 2016*; *Turtle et al., 2016*). To investigate the mechanism of how hMSC inhibited the cytotoxicity of CAR-T, the amount of CD4⁺ and CD8⁺ T cells were analyzed

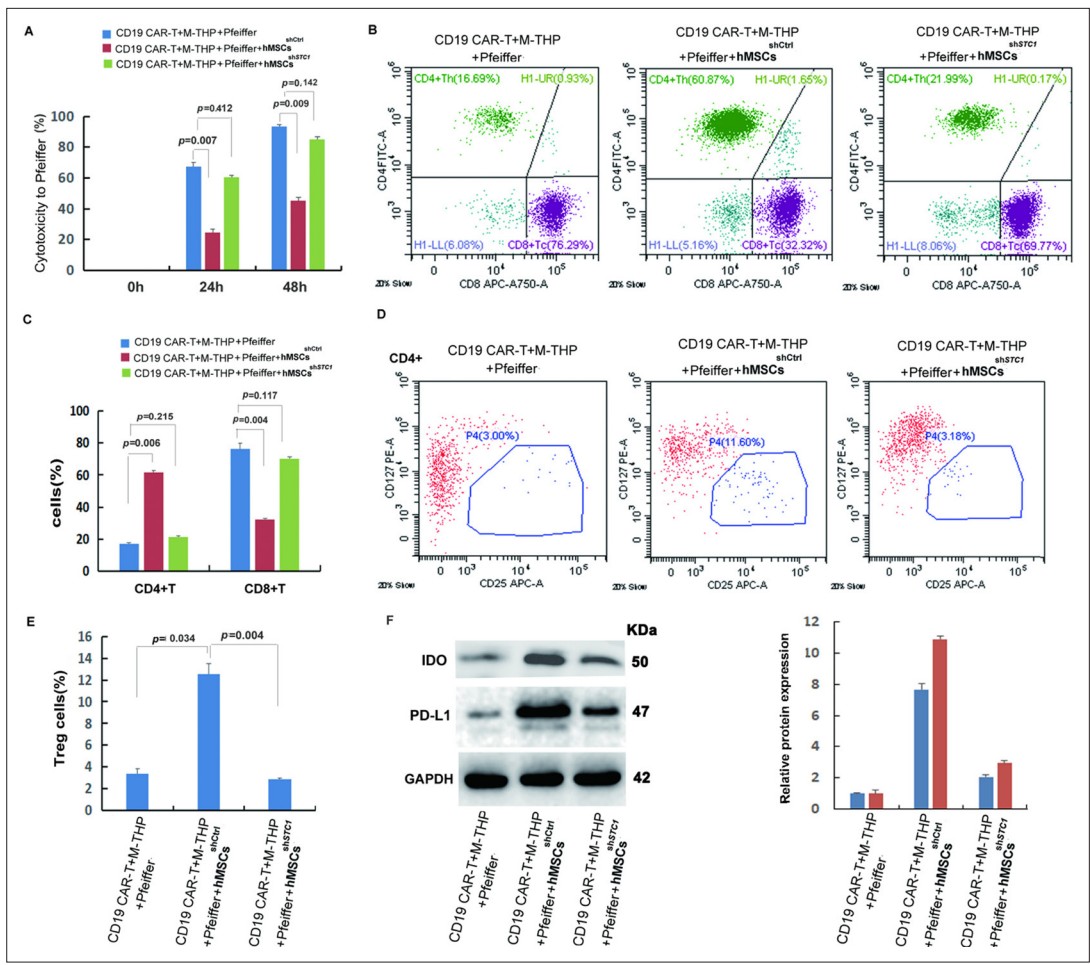

**Figure 2.** Analysis of cytotoxicity, T cell composition, and immune-suppressive markers. The cell co-culture contained chimeric antigen receptor modified T cell (CAR-T) cells, Pfeiffer cells, M2 macrophages, and control or stanniocalcin-1 (STC1) knockdown hMSCs in a ratio of 1:3:1:1. After 24 hr (or 48 hr for cytotoxicity) incubation, the following analysis was conducted: (**A**) The impact of hMSC (w/o STC1) on the cytotoxicity of CAR-T toward Pfeiffer cells; (**B**) FACS analysis of CD4$^+$ and CD8$^+$ composition. (**C**) Quantitation of the FACS data on CD4$^+$ and CD8$^+$; (**D**) FACS analysis of Treg$^+$ cells (CD4$^+$CD127$^+$CD25$^+$); (**E**) Quantitation of Treg$^+$ cells. (**F**) Western blot analysis of indoleamine 2,3-dioxygenase (IDO) and programmed cell death-ligand 1 (PD-L1) expression in the cell co-culture. Data in bar graphs are presented as the mean ± SD from three independent experiments (p values are as indicated, n=3).

The online version of this article includes the following source data for figure 2:

**Source data 1.** *Figure 2A* in Excel file.

**Source data 2.** *Figure 2C* in Excel file.

**Source data 3.** *Figure 2E* in Excel file.

**Source data 4.** Labeled original blots of *Figure 2F*.

**Source data 5.** Unlabeled original blots of *Figure 2F*.

by flow cytometry 24 hr after co-culture. As shown in *Figure 2B and C*, the ratio between CD4$^+$ and CD8$^+$ was about 1:4 when there were no hMSCs in co-culture (*Figure 2C*). However, the addition of hMSC caused a significant increase of CD4$^+$ and a decrease of CD8$^+$ T cells (*Figure 2B*), resulting in a ratio change to 2:1. Similar to the change of CD4$^+$ T cells, the percentage of regulatory T cells (Treg) was also significantly increased from ~3 to 12% when co-culture with hMSC (*Figure 2D and E*). When using hMSC$^{shSTC1}$, all the changes were completely reversed back to the level similar to that of co-culture without hMSCs. This explains the reduced CAR-T cytotoxicity since CD8$^+$ T cells are directly responsible for specific lytic activity against lymphoma (*Sommermeyer et al., 2016*). Tregs, which

account for 5–10% of the total number of CD4[+] T cells, are known to play a role in suppressing the function of T cells and other immune cells (*Zhang et al., 2018*). Therefore, the above results indicate that hMSCs' inhibitory effect on CAR-T cytotoxicity was due to both suppression of CD8[+] cells and the induction of Treg cells, and the presence of STC1 was indispensable for these impacts of hMSC.

## The presence of hMSC enhanced immune suppression and STC1 played a key role

The immune-suppressive TME is the main cause of CAR-T cell exhaustion which attenuates its efficacy. To further investigate the function of STC1 and the molecular mechanism of hMSC on CAR-T resistance, some key regulators of TME were determined. As shown in *Figure 2F*, the addition of hMSC to the cell co-culture stimulated the expression of indoleamine 2,3-dioxygenase (IDO) and programmed cell death-ligand 1 (PD-L1). IDO and PD-L1 are two of the most important immunosuppressive proteins. IDO is an intracellular enzyme that converts tryptophan into inhibitory metabolites for T-cell activity (*Ninomiya et al., 2015*). PD-L1 is expressed in tumor cells and immune cells contributing to the immune-suppressive TME (*Ribas and Hu-Lieskovan, 2016*). When using hMSC[shSTC1], the expression level of IDO and PD-L1 was both significantly reduced by more than 50%, though still higher than that without hMSC. These results indicated that the presence of hMSC can enhance the expression of immune suppressive proteins in Pfeiffer cells and macrophages, and the presence of STC1 is important for hMSC to exert these effects.

## hMSCs suppressed key components of the NLRP3 inflammasome by modulating mitochondrial ROS release

In the co-culture model, M2 macrophages were included since a previous study showed that macrophages could activate hMSCs to secrete STC1 (*Cen et al., 2019*). In addition, the macrophage is a critical part of immune response and an important regulator of immunotherapy (*DeNardo and Ruffell, 2019*). To further identify the mechanisms mediating the inhibitory effects of hMSCs, the activation of the NLRP3 inflammasome was determined. The NLRP3 inflammasome is a critical component of the innate immune system mediating caspase-1 activation and proinflammatory cytokines secretion in response to harmful stimuli such as infection and endogenous stress (*Menu and Vince, 2011*). As shown in *Figure 3A*, the release of cleaved caspase-1 p20 in cell lysates, which is the indicator of caspase-1 activation, was detected after the PMA polarization of THP-1 cells to form the M1 macrophages (M-THP1). Following co-culture with CD19 CAR-T, the level of cleaved caspase-1 was significantly upregulated. The increase of active caspase-1 was abrogated when hMSCs were added into the co-culture. knock-down of STC1 led to another reverse and completely blocked the inhibitory function of hMSCs (*Figure 3A*). Concomitant with the reduction in active caspase-1, the cleaved IL-1β mature form and absent-in-melanoma 2 (AIM2), two key components of the inflammasome (*Kelley et al., 2019*), were both increasingly expressed following M-THP1 polarization and further incubation with CAR-T (*Figure 3A*). Compared to the partial inhibition of the active caspase-1 formation, the addition of hMSC in the cell co-culture showed a stronger inhibition of these two proteins, and their expression level was returned to the base level of Pfeiffer plus CAR-T (*Figure 3A*). This result suggests that the immune-suppressive effect of hMSC was through its impact on macrophages, not CAR-T or Pfeiffer cells. Knockdown of STC1 abrogated the inhibition of hMSC on IL-1β and AIM2 (*Figure 3A*). The levels of IL-1β in the supernatants measured by ELISA showed similar results as cell lysate (*Figure 3B*).

Mitochondrial dysfunction is one of the major stimuli that activates the NLRP3 inflammasome, and it was reported that exogenous STC1 is internalized by macrophages within 10 min and localizes to mitochondria to suppress superoxide generation (*Wang et al., 2009*). Therefore, we determined the impact of hMSC on the intracellular level of reactive oxygen species (ROS) and mitochondria mass in macrophages by fluorescent dye CellROX and MitoTracker Green, respectively. As shown in *Figure 3C and D*, the presence of hMSCs[shCtrl] markedly suppressed both the cellular and mitochondrial ROS induced by the co-culture of CAR-T cells, tumor cells, and macrophages. Knockdown of STC1 eliminated the function of hMSC in suppressing ROS. This result correlates well with the expression of caspase-1, IL-1β, and AIM, suggesting that hMSCs inhibited NLRP3 inflammasome activation in macrophages was most likely by inhibiting the oxidative burst.

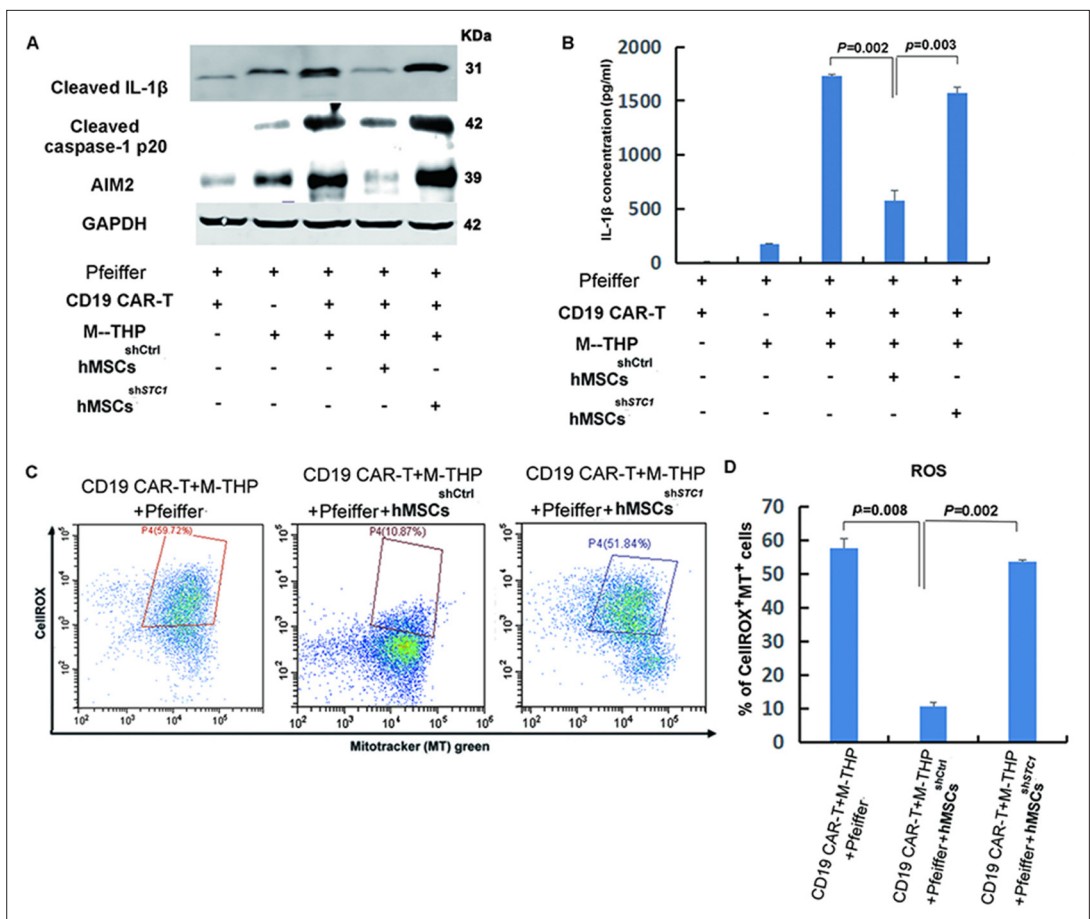

**Figure 3.** The impact of mesenchymal stem cells (MSCs) on the expression of key components involved in the formation of NLRP3 inflammasome and mitochondrial reactive oxygen species (ROS). (**A**) The protein expression of IL-1β, caspase-1, and AIM2 in cell lysates was analyzed by Western blot. (**B**) Quantitation of IL-1β secretion in the supernatants by ELISA. (**C**) FACS analysis of ROS level and mitochondria mass with fluorescent dye CellROX Deep Red and MitoTracker Green. (**D**) Quantitation of mitochondria-specific ROS level based on the percentage of cells that were both positive for CellROX and MitoTracker. All samples were collected 24 hr post the co-culture of different cells. For the measurements of IL-β, results are shown as the mean ± SD from three independent experiments (p values are as indicated, n=3).

The online version of this article includes the following source data for figure 3:

**Source data 1.** Labeled original blots of *Figure 3A*.

**Source data 2.** Unlabeled original blots of *Figure 3A*.

**Source data 3.** *Figure 3B* in Excel file.

**Source data 4.** *Figure 3D* in Excel file.

## hMSCs showed strong inhibition on CD19 CAR-T therapy in xenograft mice, which was abrogated by STC1 knockdown

The immune-suppressive impact of hMSC on CAR-T therapy and the function of STC1 were further evaluated in a xenograft model. Upon injection of Pfeiffer cells and confirmation of engraftment, we injected hMSC into the tumor area while applying CAR-T treatment by tail vein injection. As shown in *Figure 4A*, CD19 CAR-T treatment combined with the injection of hMSC[shSTC1] achieved a significant curative effect, and the tumors nearly disappeared at day 38. However, the hMSC[shCtrl] group showed a continued increase in tumor size and spreading of tumor.

Based on the immunohistochemical analysis of IL-1β in tumor tissue on day 10, the number of positive cells (brownish-yellow staining) ranged from 76 to 100% in the hMSC[shSTC1] group, while it ranged from 5 to 20% in the hMSC[shCtrl] group, indicating that hMSC could suppress TME and STC1

knockdown significantly diminished this impact (*Figure 4B*). Consistent with the results in vitro, a large amount of CD4$^+$ T cells were detected in the hMSC$^{shCtrl}$ group but much less in the hMSC$^{shSTC1}$ group. On the contrary, the amount of CD8$^+$ T cells was significantly increased in the hMSC$^{shSTC1}$ group compared to that of the hMSC$^{shCtrl}$ group (*Figure 4B*). Based on the staining of FOXP3 (forkhead box P3), a master regulator involved in the development of Treg cells, the amount of Treg cells was also evidently increased in the hMSC$^{shCtrl}$ group compared to that of the hMSC$^{shSTC1}$ group (*Figure 4B*). These results further confirmed that knockdown of STC1 abrogated the immune-suppressive capability of MSC.

The changes in the average radiance were consistent with the changes in the tumor size (*Figure 4C and D*). The survival time of mice demonstrated that mice in CAR-T combined with the hMSC$^{shSTC1}$ group had the longest survival with no death by day 38 (*Figure 4D*). Compared to the control group with no CAR-T treatment, tumor spreading in the hMSC$^{shCtrl}$ group was slower, and all survived for 6 days more. These results confirmed the inhibitory effects of hMSC on CAR-T therapy under in vivo situations and demonstrated that STC1 is an important factor affecting therapy efficacy.

## Discussion

Stem cells are believed to play critical roles in resistance to cancer therapy, which is a major contributor to poor treatment responses and tumor relapse. Previous studies have been mainly focused on the role of cancer stem cells. In the current study, we presented evidences that the presence of MSCs in TME may also be an important source of cancer treatment resistance. By modulating TME, MSCs showed a strong suppressive function on CAR-T efficacy toward lymphoma cells, and interestingly, the presence of the STC1 gene played a critical role.

The role of STC1 in cancer is paradoxical. Some reports showed that it exerts an oncogenic role, whereas other studies suggested the opposite (*Chen et al., 2019*). The aberrant expression of STC1 has been reported to impact various types of cancer, such as triggering tumor angiogenesis by upregulating the expression of VEGF in gastric cancer cells (*He et al., 2011*), causing tumorigenesis and poor clinical outcomes in ovarian, colorectal, and lung cancers (*Yeung et al., 2012*; *Chen et al., 2019*). To date, the potential roles of STC1 in immunotherapy are still largely unknown. Here, we demonstrated that the presence of STC1 is critical for MSC to exert its immunosuppressive role by inhibiting cytotoxic T cell subsets, activating some key immune suppressive/escape mechanisms, and crosstalk with other immune cells.

First, a significant downregulation of CD8$^+$ T Cells together with the upregulation of CD4$^+$ T helper cell subsets and Tregs indicated that the suppressed CAR-T efficacy was at least partially associated with MSC's function in modulating the proliferation of different T-cell subsets. Since the suppression of CD8$^+$ T cells was completely abrogated if knockdown STC1 in MSCs, it is clear that STC1 played a key role here. Moreover, considering that STC1 is secreted into the extracellular matrix in a paracrine manner, MSCs' modulation of the T cell subsets is most likely indirectly via altered cytokine expression or other secondary molecules activated by STC1. In line with our study, it was recently reported that STC-1 negatively correlates with immunotherapy efficacy and T cell activation by trapping calreticulin, which abrogates membrane calreticulin-directed antigen presentation function and phagocytosis (*Lin et al., 2021*).

The presence of MSCs also stimulated the expression of IDO and PD-L1, two important immune-suppressive molecules. Upregulation of IDO is an endogenous feedback mechanism controlling excessive immune responses, which can be produced both by tumor cells and macrophages (*Uyttenhove et al., 2003*). IDO-mediated formation of immunosuppressive metabolites can inhibit T-cell proliferation and induce T-cell death through the dioxin receptor (*Opitz et al., 2011*; *Frumento et al., 2002*). PD-L1 is a well-characterized molecule of the major escape mechanism of immunotherapy by inhibiting PD-1-mediated effector T cell function and downregulating antigen tolerance (*Ribas and Hu-Lieskovan, 2016*). There have been numerous studies reporting the bidirectional interactions between MSCs and cancer cells, resulting in regulating the expression of PD-L1 on the surface of various cancer cells or TME (*Aboulkheyr and Bigdeli, 2022*; *Krueger et al., 2019*; *O'Malley et al., 2018*; *Sun et al., 2018*). Importantly, here we demonstrated that the upregulated expression of both IDO and PD-L1 by MSCs was much reduced if the STC1 gene was knockdown.

The paracrine activity of MSCs is now widely recognized as an important cellular mechanism to communicate with immune cells and various other cell types in TME (*Teixeira et al., 2013*). Consistent

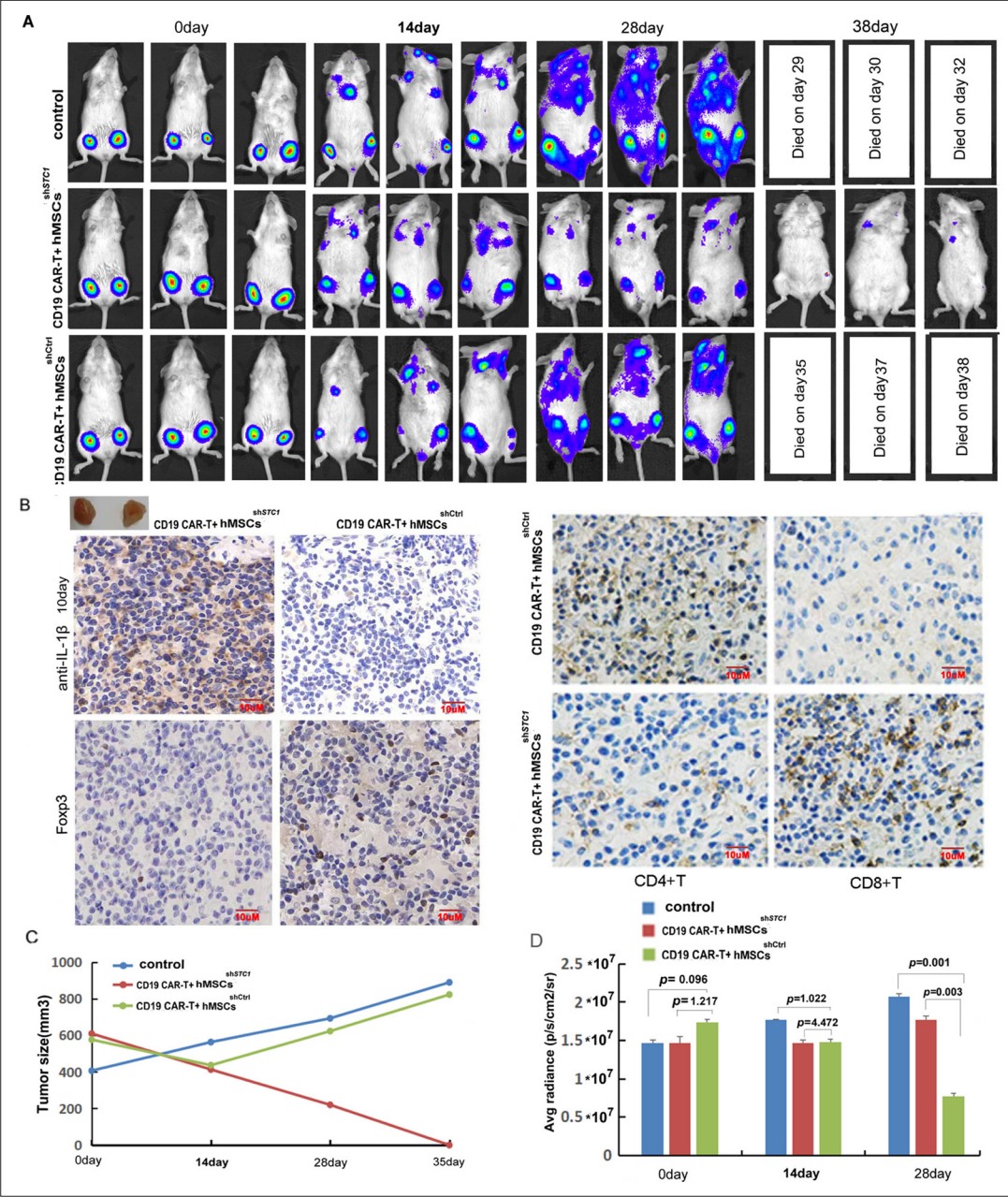

**Figure 4.** The inhibition of hMSC on chimeric antigen receptor modified T cell (CAR-T) therapy in xenograft mice relied on stanniocalcin-1. (**A**) The formation and progression of tumor in three groups of mice monitored with bioluminescence imaging: the control group without any treatment, CAR-T/M-THP1/hMSCs$^{shSTC1}$ group, and CAR-T/M-THP1/hMSCs$^{shCtrl}$ group. Day 0 was set when the engraftment was confirmed after injecting the Pfeiffer cells. (**B**) Immunohistochemical analysis of IL-1β, CD4$^+$, CD8$^+$, and Treg cells (using FOXP3 as the biomarker) in tumor tissue at day 10, positive cells display brown or brownish-yellow staining color. (**C**) The tumor size change with time. (**D**) The counted average radiance, presented as the mean ± SD (p values are as indicated, n=3).

The online version of this article includes the following source data for figure 4:

**Source data 1.** *Figure 4C* in Excel file.

**Source data 2.** *Figure 4D* in Excel file.

with previous studies, we found that the addition of hMSCs to the co-culture cell model suppressed the formation of NLRP3 inflammasome in macrophages as determined by the downregulation of some key proteins including IL-1β, the activated caspase-1, and AIM. It was reported that CD4$^+$ T cells could inhibit inflammasome-mediated caspase-1 activation and IL-1β release through TNF ligands or by interferon signaling (*Guarda et al., 2009*). Therefore, the modulation of T-cell subsets and activation of the NLRP3 inflammasome by hMSC appear to be closely connected. Since NLRP3 inflammasome is a key factor in the neuroinflammation onset in CNS injuries (*Menu and Vince, 2011*), the suppression of NLRP3 inflammasome by hMSC may be potentially beneficial in reducing the exacerbated immune responses associated with CAR-T therapy.

The formation of the NLRP3 inflammasome was reported to be through NF-κB-dependent transcription of IL-1β, IL-18, and NLRP3, whereas its activation is triggered by extracellular stimuli such as lysosomal permeability, potassium efflux, and oxidative stress (*Kelley et al., 2019*). It has been proved that the expression and secretion of STC1 in multiple cell lines can be stimulated by external stimuli, including cytokines and oxidative stress (*Nguyen et al., 2009*). Considering that exogenous STC1 could be internalized by macrophages within 10 min and localizes to mitochondria and played a suppressing role in ROS generation (*Wang et al., 2009*), we speculated that the inhibition of NLRP3 inflammasome formation might be a feedback mechanism that occurred between macrophages and hMSC. It was reported that LPS-stimulated macrophages do stimulate the expression and secretion of STC1 in hMSCs (*Oh et al., 2014*). Our data further demonstrated that knockdown of STC1 deprived the function of hMSC in suppressing all the three markers used in the current study in determining NLRP3 inflammasome formation, as well as the suppression of mitochondria ROS production. These data support the idea that a feedback regulation mechanism exists between hMSC and macrophages during CAR-T therapy.

Using the Xenograft mice model, we confirmed that the tumor-killing efficacy of CAR-T could also be inhibited by hMSCs in vivo, whereas knockdown of STC1 effectively abolished the inhibition. Immunohistochemical data indicated that the downregulation of CD8$^+$ T cells, upregulation of CD4$^+$ T helper cell subsets, and Tregs were all dependent on the function of STC1. Need to note that the amount of the injected hMSCs was much higher than that of the in vivo situation. Nevertheless, the results give a clear indication that STC1 is critical for the immune-suppressive function of hMSC.

In summary, the present study revealed a significant impact of hMSC in suppressing CAR-T efficacy and provided evidence that the STC1 gene played a critical role in the regulation of various immune-suppressive mechanisms. A speculative schema of the signaling and interactions among hMSC, macrophage, CAR-T, and tumor cells based on our current data is shown in *Figure 5*. In this model, activated macrophages or stress signals during CAR-T therapy may prompt MSCs to secret STC-1 into the extracellular matrix of TME, serving as a pleiotropic factor to negatively impact the function of T cells and stimulate the expression of molecules that inactivate immune responses, ultimately providing an immunosuppressive effect of MSC. While further studies are needed to understand the detailed molecular interactions underlying, the findings we presented here are no doubt that would have potential clinical applications toward improving the efficiency of CAR-T therapy as well as reducing the excessive toxicity by modulating the level of STC1 in TME.

## Materials and methods
### Cell culture and isolation of primary cells
HEK-293T, Peiffer, and THP-1 cells were obtained from the American Type Culture Collection (Manassas, VA, USA). The cell lines were tested mycoplasma negative using a Mycoplasma Stain Assay Kit. HEK-293T was grown in Dulbecco's Modified Eagle Medium (DMEM, Gibco) supplemented with 10% Fetal calf serum (FCS, Gibco). Peiffer cells were grown in RPMI 1640 medium supplemented with 10% FCS. Human umbilical cord blood-derived MSCs were established from consenting mothers and processed within the optimal period of 6 hr as described (*Qiao et al., 2008*), isolated cells were confirmed by surface antigen markers with flow cytometry. Peripheral blood samples were obtained from healthy donors (n=3). The scFv targeting CD19 plasmid was originated from the FMC63 clone. The CAR vectors containing scFv, human 4-1BB, and CD3z signaling domains were subcloned into the pCDHMND-MCS-T2A-Puro lentiviral plasmid. The CAR sequence was preceded by the RQR8 tag separated by a short T2A peptide for detection purposes (*Philip et al., 2014*). Ethical approval

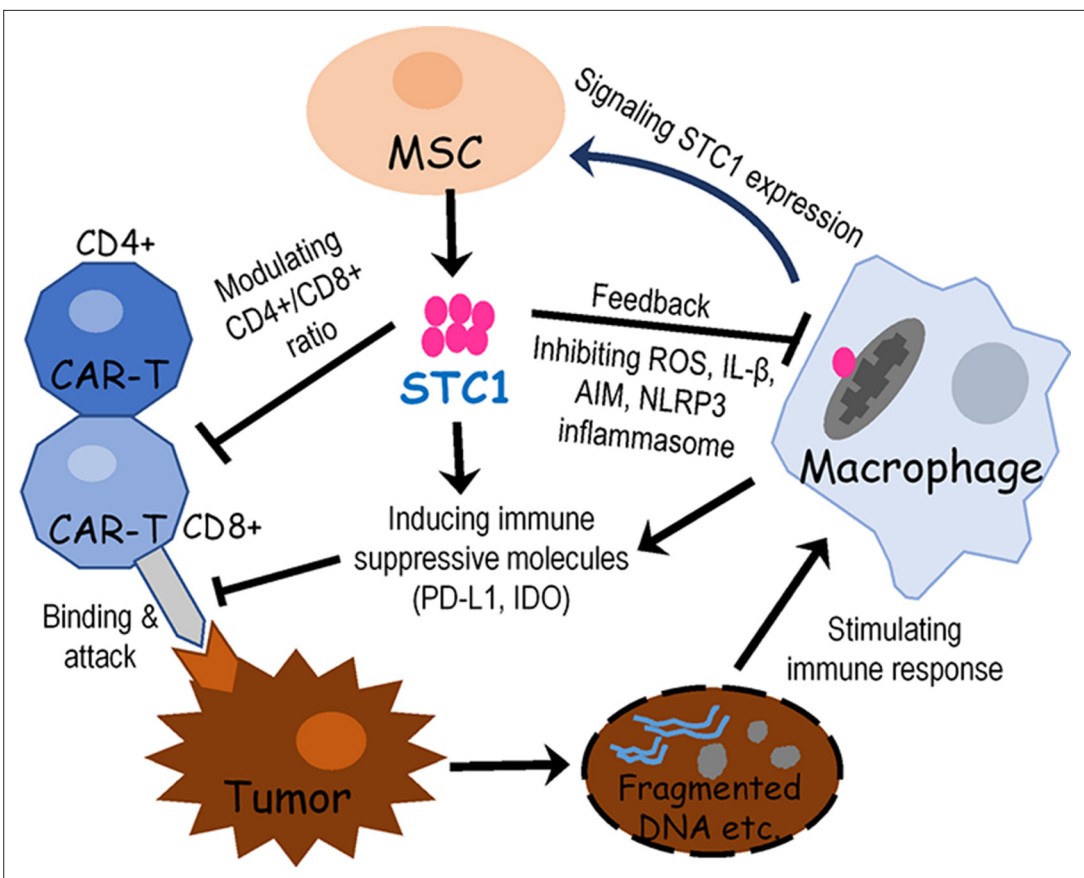

**Figure 5.** Proposed signaling and interactions among hMSC, macrophage, chimeric antigen receptor modified T cell (CAR-T), and tumor cells. When cancer cells were destroyed by CAR-T cells, the release of fragmented DNA and other stimulating factors activated the release of mitochondria reactive oxygen species (ROS) and the formation of NLPR3 inflammasome. Signals from activated macrophages and other extracellular molecules as well as oxidative stress may stimulate mesenchymal stem cell (MSC) to express and secrete stanniocalcin-1 (STC1). Then STC1 serves as a pleiotropic factor to suppress CAR-T cytotoxicity and other immune responses via direct or indirect pathways.

and informed consent were obtained in Tianjin First Central Hospital Medical Ethics Committee (Tianjin, China) for all human samples used in the current study, which was approved under clinical trial #ChiCTR-ONN-16009862.

## Lentivirus production

Preparation of the lentivirus was performed according to the manufacturer's instructions (GeneCopoeia). Briefly, HEK-293T lentiviral packaging cells in DMEM supplemented with 10% heat-inactivated fetal bovine serum (FBS) followed by transfection when cells are 70–80% confluent. Dilute 2.5 µg of lentiviral expression plasmid and 2.5 µg of Lenti-Pac HIV mix into 200 µl of Opti-MEM I (Invitrogen). In a separate tube dilute 15 µl of EndoFectin Lenti into 200 µl of Opti-MEM I, then drop-wise add to the plasmid mix and incubate for 10–25 min at room temperature. Collect the pseudovirus-containing culture medium 48 hr post-transfection followed by ultracentrifugation, and the pellets were resuspended in complete X-Vivo15 media and stored at –80°C until use.

## Production and detection of CAR-T cells

CD3$^+$ T cells from healthy donors were separated from PBMCs using CD3 immunomagnetic beads (#130-097-043, Miltenyi Biotec, Germany), then amplified using CD3/CD28 stimulation beads (#11131D, Thermo Fisher Scientific) and IL-2 (100 IU/mL; Miltenyi Biotec) in X-VIVO 15 medium (Lonza). Cells were activated and expanded for 48 hr followed by transduction 2 hr later with lentivirus. T cells

were generally engineered for 9–12 days to express a CD19-specific CAR and stained with Alexa-Fluor 647-labeled polyclonal goat anti-mouse IgG (H+L) antibodies (Affinity) to detect CAR-T cells. All cells were further confirmed by staining with fluorescein isothiocyanate (FITC)-labeled anti-CD3 antibodies (Abcam).

## Cell co-culture model

THP-1 cells ($5\times10^5$/well) were seeded into six-well plates and polarized into M2 macrophages by first treating with 320 nM PMA (Sigma) for 24 hr, then added 20 ng/mL IL-4 (PeproTech) and 20 ng/mL IL-13 (PeproTech) in the presence of PMA for another 24 hr to obtain M2 phenotype. The formation of M2 macrophages was validated by flow-cytometry based on the surface markers (CD11b+CD163+). After washing to remove all PMA and cytokines, $2\times10^5$ M2 macrophages were co-cultured with $2\times10^5$ CAR-T cells, $6\times10^5$ Pfeiffer cells, and $2\times10^5$ hMSCs in X-VIVO15 medium (Lonza) containing IL-2(100 IU/mL, MiltenyiBiotec) for 24 or 48 hr.

## Generation of STC1 knockdown cells

Lentiviral particles PLKO.1 and PLKO.1-sh*STC1* were provided by the Beijing Institute of Radiation Medicine. Viruses were packaged by co-transfection with PLKO.1 and PLKO.1-sh*STC1* into 293T cells. The supernatants containing viruses were collected 48 hr after transfection, then the centrifuged and resuspended lentivirus were used for further transduction of hMSCs in Opti-MEM. The stable STC1 knockdown hMSCs were obtained after 7–10 days of puromycin selection in 96-well plates. Transduction efficiency was determined by fluorescent microscopy.

## MTT assay

Cell viability was examined by 3-(4,5-dimethylthiazol-2-yl)–2,5-diphenyltetrazolium (MTT) assay (Sigma). The absorbance was measured using a Synergy 4 plate reader (Bioteck) with a test wavelength at 490 nm and a reference wavelength at 630 nm.

## Cell migration determination by wound healing and transwell chamber assay

hMSCs$^{shCtrl}$ and hMSCs$^{shSTC1}$ were grown on six-well plates and wounded using a sterile pipette tip. The progress of migration was recorded immediately following injury, and photo-micrographs were taken at zero and 48 hr.

For transwell assay, hMSCs$^{shCtrl}$ and hMSCs$^{shSTC1}$ were seeded into the upper chamber of a transwell cell culture insert with $1.0\times10^4$ cells in 200 μL of a 1% FBS-containing medium. The lower chamber was filled with 600 μL of medium containing 10% FBS. Twenty-four hours later, cells that had migrated to the lower side of the membrane were fixed in 4% paraformaldehyde and stained with DAPI. The migrated cells were counted and photographed in five fields of view and were done in three independent experiments.

## Apoptosis detection with annexin V-FITC and PI and TUNEL assay

An increase in the plasma membrane PS externalization occurs early in apoptosis and can be detected by annexin V staining. hMSCs$^{shCtrl}$ and hMSCs$^{shSTC1}$ were isolated and stained with annexin V-FITC and PI (Invitrogen), then apoptosis-positive cells were analyzed using FACS (Millipore Muse).

The terminal deoxynucleotidyl transferase-mediated dUTP nick-end labeling (TUNEL) assay was used to monitor the extent of DNA fragmentation as a measure of apoptosis (*Latha et al., 2005*). After hMSCs$^{shCtrl}$ and hMSCs$^{shSTC1}$ were fixed by formaldehyde, immunohistochemical detection of apoptotic cells was carried out using DeadEnd Fluorometric TUNEL System (Promega). The cells were washed with PBS and blocked with 10% goat serum, then used DAPI to stain nuclei. The samples were photographed with a confocal laser microscope (Olympus), and TUNEL-positive cells were quantitated.

## Quantitative real-time PCR

Total RNA was extracted using TRIzol reagent (Invitrogen), serving as a template for real-time PCR using random primers and M-MLV reverse transcriptase. The primers used were as follows: human TSP1: forward: 5'-TTGTTAAGAGGTTTGAG TAGGAGAG-3' and reverse: 5'-CCCACCTTACTTACCTA AAATCACA-3'.

## Western blotting and cytokine release analysis

Western immunoblotting was performed as previously described (*Zhang et al., 2016*). After SDS-PAGE and blotting, proteins were detected using the following antibodies: rabbit anti- IL-1β (Abcam, ab9722), anti-Caspase-1 p20 (Bioss, bs-10442R), AIM2 (Abcam, ab93015), IDO (Bioss, bs-15493R), PD-L1 (Bioss, bsm-54472R), and mouse anti-GAPDH (Santa Cruz) primary antibodies. The secondary antibodies were IRDye-800-conjugated anti-mouse and anti-rabbit immunoglobulin G (Li-COR Biosciences) (1:200). Immunofluorescence was detected using Odyssey Infrared Imaging System (Gene Company Ltd.). GAPDH expression was used as an internal control. The relative quantification of protein expression was analyzed using ImageJ software. The level of IL-1β in the serum was detected using ELISA by electrochemiluminescence (R&D Systems, France).

## Flow cytometry

The expression of CD4, CD8, CD127, and CD25 in CAR-T cells was analyzed using flow cytometry with the following fluorochrome-conjugated monoclonal/polyclonal antibodies (all from Caprico Biotechnologies): anti-human CD4 (CD004210403), anti-human CD8 (CD008210301), anti-human CD127 (CD127210501), and anti-human CD25 (CD025210301).

## In vitro analysis of CAR-T cytotoxicity toward Pfeiffer cells

Seeding CD19 CAR-T cells ($4×10^5$ cells/group) in a co-culture with Pfeiffer cells and macrophages polarized from M-THP1 at a 1:3:1 ratio and incubate for 48 hr. The cell killing of CAR-T toward Pfeiffer cells was determined using a LDH cytotoxicity test kit (Dojindo Molecular Technologies, Inc) and measured at 0, 24, and 48 hr after cell co-culture.

## Cellular and mitochondrial ROS detection

ROS was measured using CellROX Deep Red Reagent (Invitrogen) and MitoTracker Green FM Dye (Invitrogen) (*Lagadinou et al., 2013*; *Minai et al., 2013*). Briefly, cells were co-cultured for 24 hr followed by loading with CellROX dye (5 mM) and MitoTracker Green dye (100 nM) at 37°C for 30 min, then analyzed by flow cytometry. The data were analyzed using Flowjo software (Tree Star Inc, Ashland, OR).

## Xenograft tumor model

Female 6–8-week-old NOD/Shi-scid IL-2Rγ(null) (NOG) mice weighing 20±1.6 g (n=36, Vitonlihua Experimental Animal Technology Co., Ltd, Beijing, China) were injected with $5×10^6$ Pfeiffer cells expressing luciferase by subcutaneous injection on each side. Established tumors were monitored by bioluminescence imaging (BLI). Upon confirmation of engraftment after 25 days, the mice were randomized into three groups and treated by tail vein injection of $5×10^6$ CD19 CAR-T cells and $2.5×10^6$ M-THP1. At the same time, $5×10^6$ cells/mice of hMSCs[shSTC1] or hMSCs[shCtrl] were injected into multi-points of the tumor area. Tumor progression was photographed with BLI following intraperitoneal injection with D-luciferin (Goldbio, 150 mg/kg) at 14, 28, and 38 days. All the mice were sacrificed when either experimental or humane endpoints were reached. All animal experiments and procedures were approved by the Ethics Committee of Tianjin First Central Hospital (Tianjin, China. #2021-SYDWLL-000301).

## Immunohistochemical analysis of IL-1β, CD4[+], CD8[+], and Treg cells in vivo

Mice were sacrificed on day 10 after CAR-T/M-THP1 and hMSC injection, and tumor samples were fixed with formalin and embedded in paraffin. Tumor tissues were examined by immunohistochemistry staining as previously described (*Jiang et al., 2018*). Briefly, the sections were exposed to 3% $H_2O_2$ in methanol after deparaffinization and rehydration and then blocked with 1% BSA for 30 min at room temperature. After blocking, the sections were incubated with primary antibodies (all from Servicebio Technology Co., China) for IL-1β (GB11113), CD4[+] (GB13064-1), CD8[+] (GB13068), and FOXP3 (GB11093) overnight at 4°C, followed by incubation with peroxidase-conjugated secondary antibodies. IL-1β+ cells were quantified by measuring the number of stained cells.

## Acknowledgements

We thank James Westmoreland and Haipei Ma for proofreading the manuscript. This work was supported by the National Key R&D Program of China (2018YFA0901702), the Shandong Key R&D Program (2019GSF107088), and the National Science Foundation of Shandong (ZR2020MC077, ZR202111220001).

## Additional information

### Funding

| Funder | Grant reference number | Author |
| --- | --- | --- |
| National Key Research and Development Program of China | 2018YFA0901702 | Wenjian Ma |
| Shandong Key Research and Development Program | 2019GSF107088 | Qingxi Liu |
| National Science Foundation of Shandong | ZR202111220001 | Wenjian Ma |
| National Science Foundation of Shandong | ZR2020MC077 | Wenjian Ma |

The funders had no role in study design, data collection and interpretation, or the decision to submit the work for publication.

### Author contributions

Rui Zhang, Conceptualization, Formal analysis, Methodology, Writing – original draft; Qingxi Liu, Formal analysis, Methodology; Sa Zhou, Methodology; Hongpeng He, Formal analysis; Mingfeng Zhao, Resources, Supervision; Wenjian Ma, Conceptualization, Writing - review and editing

### Author ORCIDs

Wenjian Ma http://orcid.org/0000-0002-3392-1549

### Ethics

Human subjects: Ethical approval and informed consent were obtained. Patients with lymphoma and Healthy donors agreed to participate in this experiment within a clinical trial at the Department of Hematology at Tianjin First Central Hospital (Tianjin, China) with autologous CAR-T 19 cells (ChiCTR-ONN-16009862; Tianjin First Central Hospital Medical Ethics Committee) in accordance with the World Medical Association medical research guidelines. Peripheral blood samples were obtained from healthy male donors (n = 3) in Tianjin First Central Hospital.

Animal experimentation: This study was performed in strict accordance with the recommendations in the Guide for the Care and Use of Laboratory Animals of the National Institutes of Health. All animal experiments and procedures were approved by the Ethics Committee of Tianjin First Central Hospital (approval#2021-SYDWLL-000301).

### Decision letter and Author response

Decision letter https://doi.org/10.7554/eLife.82934.sa1
Author response https://doi.org/10.7554/eLife.82934.sa2

## Additional files

### Supplementary files

• MDAR checklist

### Data availability

All data generated or analysed during this study are included in the manuscript. Source data files have been provided for Figures 1, 2, 3 and 4.

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
