## [Editor Report]

This study uncovers the contributions of MSC on modulating CAR T-cell behaviour. Based on the importance in basic biology and its immediate impact on translational potential, all reviewers are satisfied on the advances in this study.

---

## [Decision Letter]

**Decision letter after peer review:**

Thank you for submitting your article "Mesenchymal stem cell suppresses the efficacy of CAR-T toward killing lymphoma cells by modulating the microenvironment through stanniocalcin-1" for consideration by eLife. Your article has been reviewed by 2 peer reviewers, one of whom is a member of our board of Reviewing Editors, and the evaluation has been overseen by Tadatsugu Taniguchi as the Senior Editor. The reviewers have opted to remain anonymous.

Essential revisions:

The mechanistic understanding will be improved if new data can be included. Please pay attention to the following points.

1) How STC1 controls changes in MSCs' ability for hampering CAR T cell-mediated anti-tumor responses is unclear.

2) What does the author think about using MSC with STC-1 knockout? Can it still help reduce toxicity while maintaining CAR-T efficacy? This might be a potential application.

*Reviewer #1 (Recommendations for the authors):*

Overall, the mechanistic part of this manuscript should be largely improved. Moreover, a careful evaluation of how MSCs modulate CAR T cell properties and differentiation should be examined. In addition, the methodology part should be elaborated since it is vague in many experimental settings.

*Reviewer #2 (Recommendations for the authors):*

1. Were all experiments repeated at least three times? This info should be stated in methods or figure legend. statistical significance should be added to Figure 1B and figure 4D.

2. Page 8, how was CAR-T cytotoxicity determined, isolated Pfeiffer cell or the total cell co-culture? Similarly, was ROS determined specifically in macrophage, or the cell co-culture?

3. The fonts in Figure 1C are rather small, and should be enlarged.

4. What are the green dots in Figure 1 G's right panel? There should be some explanation either in legend or text.

5. Quantitation of protein expression level should be added in Figure 2F western blot.

---

## [Author Response]

Essential revisions:The mechanistic understanding will be improved if new data can be included. Please pay attention to the following points.1) How STC1 controls changes in MSCs' ability for hampering CAR T cell-mediated anti-tumor responses is unclear.

In this study, we demonstrated that the presence of STC1 is critical for MSCs to exert their immunosuppressive role by inhibiting cytotoxic T cell subsets, activating key immune suppressive/escape related molecules such as IDO and PD-L1, and crosstalking with macrophages in the TME. These immunosuppressive functions of MSC could be significantly hampered when the STC1 gene was knockdown. Considering that staniocalcin-1 is glycoprotein hormone that is secreted into the extracellular matrix in a paracrine manner, we would conclude that the role of STC-1 is not to alter the function of MSCs intracellularly. Rather, it facilitates the immunosuppressive capabilities of MSCs through extracellular secretion into the TME as a pleiotropic factor, thus impacting the functioning of T cells, cancer cells and other immune cells.

The reviewer's question is well taken, and we have added the points mentioned above to the Discussion section to ensure a more comprehensive conclusion. Moreover, a recent study published in Cancer Cell, which was suggested by the other reviewer, is consistent with our results. It has provided further mechanistic information on how stanniocalcin-1 impacts immunotherapy efﬁcacy and T cell activation. The reference has been cited and discussed as shown below.

"In this model, activated macrophages or stress signals during CAR-T therapy may prompt MSCs to secret staniocalcin-1 into the extracellular matrix of TME, serving as a pleiotropic factor to negatively impact the function of T cells and stimulate the expression of molecules that inactivate immune responses, ultimately providing an immunosuppressive effect of MSC.".

"In line with our study, it was recently reported that stanniocalcin-1 negatively correlates with immunotherapy efﬁcacy and T cell activation by trapping calreticulin, which abrogates membrane calreticulin-directed antigen presentation function and phagocytosis [50]."

2) What does the author think about using MSC with STC-1 knockout? Can it still help reduce toxicity while maintaining CAR-T efficacy? This might be a potential application.

This is definitely an interesting idea. Based on the data presented in the current study, it is clear that knockdown of STC-1 would abrogate the immune-suppressive impact of MSC, and therefore affect CAR-T efficacy. However, whether the presence of MSC can help reduce cytokine release syndrome when losing the function of STC-1 requires further study. We agree with the reviewer, and we had briefly discussed this possibility at the very end of the discussion as shown below.

"… the findings we presented here are no doubt that would have potential clinical applications toward improving the efficiency of CAR-T therapy as well as reducing the excessive toxicity by modulating the level of STC1 in TME".

Reviewer #1 (Recommendations for the authors):Overall, the mechanistic part of this manuscript should be largely improved. Moreover, a careful evaluation of how MSCs modulate CAR T cell properties and differentiation should be examined. In addition, the methodology part should be elaborated since it is vague in many experimental settings.

We have revised the introduction and discussion to include more comprehensive information and analysis of the underlying mechanisms.

As to the methodology, we have conducted a thorough examination of the methodology section, providing more in-depth details, such as the generation of M2 stage macrophages and the cell co-culture model.

Reviewer #2 (Recommendations for the authors):1. Were all experiments repeated at least three times? This info should be stated in methods or figure legend. statistical significance should be added to Figure 1B and figure 4D.

Yes, all experiments were sufficiently repeated. This info is now added in multiple places of the Figure legends. Statistical significance was added to Figure 1B and figure 4D.

2. Page 8, how was CAR-T cytotoxicity determined, isolated Pfeiffer cell or the total cell co-culture? Similarly, was ROS determined specifically in macrophage, or the cell co-culture?

Both experiments were done on total cell co-culture, it was not possible to isolate the single cell line. However, by comparison with control, the results gave reliable determination of cytotoxicity and the changes in ROS levels. This information is added to Results (page 13).

3. The fonts in Figure 1C are rather small, and should be enlarged.

Done as suggested.

4. What are the green dots in Figure 1 G's right panel? There should be some explanation either in legend or text.

The green dots were from the background. The images were a bit overexposed in order to demonstrate that knockdown of STC-1 does not cause apoptosis. This explanation was added to Results.

5. Quantitation of protein expression level should be added in Figure 2F western blot.

Done as suggested.